# Cytokine Expression Patterns and Single Nucleotide Polymorphisms (SNPs) in Patients with Chronic Borreliosis

**DOI:** 10.3390/antibiotics8030107

**Published:** 2019-07-30

**Authors:** Tabea M. Hein, Philip Sander, Anwar Giryes, Jan-Olaf Reinhardt, Josef Hoegel, E. Marion Schneider

**Affiliations:** 1Division of Experimental Anesthesiology, University Hospital Ulm, Albert-Einstein-Allee 23, 89081 Ulm, Germany; 2Swiss Medical Clinic AG, Grütstrasse 55, CH-8802 Kilchberg, Switzerland; 3Practice for General Medicine, Sports Medicine, and Endocrinology, Leopoldstraße 17, 75172 Pforzheim, Germany; 4Institute of Human Genetics, Ulm University, Albert-Einstein-Allee 11, D-89081 Ulm, Germany

**Keywords:** Lyme borreliosis, CD163, IL-1β, IL-6, IL-8, TNF-α, pro-inflammatory cytokines, IL-6 promotor SNP rs1800795, persisters, pathogens

## Abstract

(1) Background: Genetically based hyperinflammation may play a role in pathogen defense. We here questioned whether alterations in circulating monocytes/macrophages, inflammatory biomarkers and a functional SNP (single nucleotide polymorphisms) of the Interleukin-6 (IL-6) promotor might play a role in patients with persistent, and treatment resistant borreliosis. (2) Methods: Leukocyte subpopulations were studied by flow cytometry; plasma cytokines were determined by a chemiluminescence based ELISA (Immulite^®^), and genotypes of the IL-6 promotor SNP rs1800795 were determined by pyrosequencing. (3) Results: In a cohort of *n* = 107 Lyme borreliosis patients, who concomitantly manifested either malignant diseases (group 1), autoimmune disorders (group 2), neurological diseases (group 3), or morbidities caused by multiple other infectious complications (group 4), we found decreased numbers of anti-inflammatory CD163-positive macrophages, elevated concentrations of inflammatory cytokines, and an imbalance of IL-6 promotor SNP rs1800795 genotypes. The most prominently upregulated cytokines were IL-1β, and IL-8. (4) Conclusions: Increased pro-inflammatory phenotypes identified by monocyte/macrophage subtypes and concomitantly increased cytokines appear to be valid to monitor disease activity in patients with persistent Lyme borreliosis. Patterns may vary by additional co-morbidities. In patients with autoimmune diseases, increased frequencies of a heterozygous IL-6 promotor SNP rs1800795 were identified. This functional SNP may guide chronic inflammation, impacting other cytokines to trigger trigger chronicity and therapeutic resistance in Lyme borreliosis.

## 1. Introduction

Lyme borreliosis is an emerging infectious disease that poses a global public concern as a persistent infection [1]. The causative agent *Borrelia burgdorferi* sensu lato, comprises 20 known genospecies, three of which are the primary cause of Lyme borreliosis. These genospecies occur in different geographical regions: While *B. burgdorferi* is more frequent in Northern America and Western Europe, *B. garinii* and *B. afzelii* can be found in Eurasia [1,2]. Antibiotics are the primary treatment of Lyme borreliosis, however significant improvement has been described for patients suffering from chronic Lyme borreliosis by antioxidative treatment regimen [3]. The treatment regime depends on the stage of the infection [4,5]. However, months after completion of the antibiotic therapy, ~10% of the patients report a persistence of disease-relevant symptoms like fatigue, pain, and cognitive dysfunction. These symptoms may persist for years and might even be functionally disabling (post-treatment Lyme disease syndrome) [1,6,7]. Despite antibiotic therapy, there is evidence for chronic bacterial infection by the identification of pathogens in a replicating state [8]. Other disease states may be due to intracellular round bodies [9] and immunologically inert biofilms [10].

The main clinical symptoms of acute Lyme borreliosis are caused by immune activation and the release of inflammatory cytokines. Pathogen recognition of *B. burgdorferi* occurs through pattern recognition receptors (PRR), especially Toll-like receptor 2 (TLR2) and a member of the NACHT (NAIP (NLP family apoptosis inhibitor protein), CIITA (class II major histocompatibility complex transactivator), HET-E, TEP1 (telomerase-associated protein))-LRR (Leucine-rich repeat)-protein family, NOD2 (Nucleotide-binding oligomerization domain-containing protein 2) [11,12,13]. These PRRs are central components of the inflammatory response to *Borrelia* and are able to recognize bacterial lipoproteins [14]. As a consequence, mitogen-activated protein kinase (MAPK) pathways and nuclear factor-κB (NF-κB) are upregulated [15]. Subsequently, the cytokines produced in vivo are predominantly pro-inflammatory and include tumor-necrosis-factor (TNF)-α, IL-1β, IL-2, IL-6, and IL-8 [16,17,18,19]. IL-6 is unique by its properties as an inflammatory as well as regulatory cytokine influencing IL-1, IL-8, and TNF-α [20]. Aside from upregulation of the *Borrelia*-specific TLR1/2 heterodimer [13,21] other members of the TLR family are also induced, which results in the release of type-I-interferons (IFN I) in addition [13,22,23,24]. Neutrophils are primarily and monocytes are secondarily recruited to the site of infection [25]. After a certain delay, which may last up to two days, the spirochetes start to disseminate to distant tissues. While the innate and adaptive immune mechanisms are able to reduce the pathogen load within several weeks, some spirochetes may survive and hide out in localized niches of either macrophages or collagen-rich tissues [8,26,27,28]. There, they can survive for several years through evasion of immune control, leading to the persistence of the infection-related clinical symptoms [29] causing chronic inflammation, and immune paralysis. In addition to the inflammatory clinical pathology inducing systemic pain, inflamed skin, and autoimmunity such as rheumatoid arthritis [30,31,32,33], as well as neurological dysfunction [34,35,36] may occur. Clinical phenotypes may be related to tissue preferences of *Borrelia* subspecies and as yet undefined genetic predispositions in individual patients.

The high variability in clinical symptoms and the fact that only a subpopulation of patients is apparently resistant to antibiotic therapy [37] motivated us to investigate genetic polymorphism related to the inflammatory response profile against bacterial infections of a given patient.

The IL-6 gene is a 5 kb long gene located at chromosome 7p15.3. The IL-6 promotor SNP-174G-C (rs1800795) was first described in 1998 by Fishman et al., in that the G variant (G/G and G/C) is linked to higher IL-6 secretion in vitro as compared to the C variant (C/C) [38]. This promotor SNP has since been associated with various conditions such as heart diseases [39,40], diabetes mellitus type II [41,42], and cancer [43,44]. While the G variants (G/G and G/C) but not the C variant (C/C) lead to significantly higher IL-6 levels after stimulation with LPS or IL-1 in vitro [38], one may hypothesize that many inflammatory symptoms in Lyme borreliosis are related to increased IL-6 influenced by its promotor SNP. The current study aimed at the characterization of a cohort of patients with persistent Lyme borreliosis and their inflammatory cytokine phenotype related to the presence and absence of this IL-6 promotor polymorphism. IL-6 is a unique cytokine by its potency to upregulate inflammation as well as anti-inflammatory and tissue reconstitutive pathways and antibody-production. These opposing functions of IL-6 depend on IL-6 receptors (p80 and gp130), expressed by different target tissues, and function as IL-6 buffers [45]. Basically, persistent Lyme borreliosis may occur by a detrimental effect of chronic inflammation as well as specific or generalized immune insufficiencies. This context motivated us to concentrate on genetics of the IL-6 production. We also determined other cytokines such as IL-1β, IL-8, and TNF-α on the protein level. Results may direct further studies on regulatory elements of inflammatory cytokines.

## 2. Results

### 2.1. Inflammatory Phenotypes in Patients with Persistent Lyme Disease

Among circulating leukocytes, macrophage phenotypes may mirror a current status of immune activation in a given host. In whole blood samples we identified monocytes by (i) scatter analysis and (ii) expression densities of CD45 and CD14. Monocytes identified by scatter analysis, were then tested for expression of -CD163 by using FITC-labeled antibodies. Gating strategies are explained in Appendix A. As demonstrated in Figure 1, CD14 expression is not different between healthy donors and persistent Lyme borreliosis patients (*p* = 0.37). However, the amount of CD163 positive monocytes was much lower in patients as compared to healthy controls (Figure 1). These results imply that even long-term infection does not generally affect the expression of CD14 on monocytes/macrophages in the circulation.

CD163 is a scavenger receptor expressed by anti-inflammatory, so-called M2 monocytes/macrophages. In comparison to a healthy control group with a median of 29.96%, the percentage of CD163 positive cells is significantly lower in Lyme borreliosis patients (median: 5%) (Figure 1; Mann-Whitney-U test *p* < 0.0001). The outliers may be caused by secondary disease activity. The overall decrease of M2 monocytes/macrophages support the interpretation of a chronic inflammatory milieu, counteracting tissue healing processes.

### 2.2. Inflammatory Cytokines are Increased in Persistent Lyme Borreliosis Patients

Compared to the healthy controls, Lyme borreliosis patients show an elevation of pro-inflammatory cytokines (Figure 2). IL-1β is not only an important inflammatory mediator but also plays a key role in the manifestation of pain and fever and is a product of inflammasome activation. The concentration of IL-1β is higher in Lyme borreliosis patients (Figure 2a). Long-term Lyme borreliosis patients often suffer from non-specific migrating pain, affecting varying parts of the body [46]. Therefore the elevation of IL-1β in the patient cohort indicates an ongoing inflammatory process. Differences in IL-6 (Figure 2b) are remarkable for individual patients but medians were not significantly different. TNF-α concentrations (Figure 2d) were similar in both cohorts. On the other hand, inflammatory IL-8 is more elevated in individual patients and differences are statistically different, as well (*p* < 0.0001) (Figure 2c). IL-8 is known to induce chemotaxis, neutrophil recruitment, and is increasingly secreted in response to oxidative stress. The significantly increased concentration of IL-8 in plasma samples of the patient cohort appears to be linked to elevations of IL-1β, supporting a chronic inflammatory state.

In comparison to the control group, Lyme borreliosis patients presented with higher levels of IL-1β and IL-8, with some extreme outliers in IL-6.

### 2.3. Inflammatory Cytokines Differ in Secondary Diseases of Patients with Persistent Lyme Disease

In the next step, the patient population was stratified into groups 1–4 depending on their secondary disease categories (Table 1).

More than half of the patients have, aside from Lyme borreliosis, a secondary disease. The smallest of the four subgroups suffered from malignancies: Six out of ten patients had an inflammatory mammary carcinoma, three had lung carcinoma, and one esophageal cancer (group 1). The second subgroup included patients with additional autoimmune diseases: Multiple sclerosis, Hashimoto’s thyroiditis, diabetes mellitus type I, Morbus Basedow, and arthritis (group 2) (*n* = 27; Table 1). The third subgroup consisted of patients exhibiting neurological dysfunctions including Parkinson’s disease, amyotrophic lateral sclerosis, chronic fatigue syndrome, chronic pain syndromes, neuroborreliosis, and dysesthesia (group 3; *n* = 30; Table 1). The largest subgroup included patients without secondary diseases but with multiple bacterial and/or viral coinfections (*n* = 40; Table 1; group 4).

The majority of the patients were diagnosed with a secondary condition aside from Lyme borreliosis (Table 1). The expression levels of the cytokines: IL-1β, IL-6, IL-8, and TNF-α as well as the amount of CD163+ M2 macrophages were analyzed in defined patient groups and compared to healthy controls (Table 2).

Patients with malignancies (Group 1) showed increased IL-1β, IL-6, and IL-8 as well as decreased numbers of CD163+ M2 macrophages. Group 1 is also the only one with higher plasma IL-6 when compared with the controls. Patients from the autoimmune cohort (group 2) exhibited the strongest decrease of CD163+ (M2-) macrophages as well as significantly higher IL-8 plasma concentrations. IL-1β and IL-6 on the other hand remained to be in the same range as the controls. Group 3 patients with neurological dysfunctions, and Group 4 featured with more IL-1β and IL-8. All groups experienced an obvious decrease in M2 macrophages, however, not strong enough in group 3 to attain statistical significance. Statistical tests turned out to be not significant for TNF-α comparisons for all groups of patients with persistent Lyme borreliosis.

### 2.4. Allele Distribution of IL-6 Promotor rs1800795 in Patients with Persistent Lyme Disease

A total of *n* = 107 patients were genotyped for their allele variant of the IL-6 promotor polymorphism rs1800795. A representative pyrogram for sequencing this single nucleotide polymorphism (SNP) present in wildtype (G/G), heterozygous (G/C) and homozygously (C/C) mutated gDNA is shown in Appendix A. Not contradictory to the Hardy Weinberg equilibrium (HWE; *p* = 0.104), a higher proportion of heterozygotes was observed in Lyme borreliosis cohort (55.1%) compared to the *e!Ensembl* reference population (44.9%), though not nominally significant (Table 3).

When IL-6 promotor rs1800795 genotypes in Lyme borreliosis patients were addressed according to the group designations of secondary disease phenotypes, we found that especially group 1 (70%) and group 2 (66.7%) had elevated frequencies of heterozygous (G/C) genotypes. Those were, however, again not nominally significant (Table 4). By contrast, the residual Lyme borreliosis patient groups (1, 3, and 4) were not different from the control and the *e!Ensembl* reference population, respectively.

### 2.5. Different IL-6 rs1800795 Genotypes were Associated with a Decreased Number of CD163+ Macrophages and Elevated Cytokine Levels

Patients with different IL-6 promotor genotypes were analyzed for cytokines and CD163+ M2 monocytes/macrophages. As shown in Table 5, the most remarkable finding was that IL-6 promotor SNP heterozygous individuals had fewer CD163+ anti-inflammatory macrophages. The pro-inflammatory effect of IL-6 promotor SNP-heterozygous individuals was supported by increased concentrations of IL-8 in individuals carrying this genotype. Statistical testing further supported that IL-1β and IL-6 levels were also higher in G/C heterozygous individuals (Table 5).

## 3. Discussion

Chronic inflammation plays a major role in treatment resistant infectious diseases such as Lyme borreliosis. We here addressed the phenotypes of circulating monocytes/macrophages, pro-inflammatory cytokines and a SNP genotype located in the promotor of the *IL-6* gene. Using whole blood for this analysis, there was no difference in numbers of CD14-positive monocytes between patients and controls, although CD14-positive monocytes may well increase in chronic infections [48]. However, when distinguishing functional subgroups of circulating monocytes (M1 pro-inflammatory versus M2 anti-inflammatory monocytes) [49], the majority of Lyme borreliosis patients presented with low to very low amounts of CD163+ M2 macrophages. Although not yet described in humans, *Borrelia* infected mice showed evidence for increased amounts of M1 macrophages during acute infection, but non-inflammatory immunosuppressive M2 macrophages resided to infected organs in chronic disease states [50]. Moreover, exogenous agents mediating immune insufficiency such as corticoid steroids increase the amount of *Borrelia* in an in vivo mouse model [27,50,51]. Significantly reduced amounts of CD163+ macrophages were found in Lyme borreliosis patients grouped according to the concurrent manifestation of autoimmune diseases (Table 2).

Further biomarkers in persistently infected patients were members of the pro-inflammatory biomarkers such as IL-1β, IL-6, and IL-8 but not TNF-α plasma concentrations were not different from control individuals. Amongst these, IL-8 was significantly upregulated in Lyme borreliosis patients with malignancies as secondary disease phenotype. Khazali and co-workers verified the relevance of IL-8 as a pro-inflammatory cytokine in mamma carcinoma [52]. This context may support the relevance of pathogen-related IL-8 upregulation in patients with malignancies. IL-1β as a major cytokine induced by inflammasome activation [52], was also upregulated in patients with carcinoma (Group 1) in addition to Lyme borreliosis patients with neurological diseases (Group 3), and in patients with multiple additional infections (Group 4). As studied in greater detail, inflammasome activation occurs by a range of danger signals during infections as well as tissue damage [53]. Oosting and coworkers were the first to describe the functional role of inflammasome activation by *Borrelia* related to IL-17 guided-pathology in chronic Lyme borreliosis patients in a late stage [54]. In the present report, IL-6 was not as significantly upregulated as IL-1β and IL-8 with the exception of Group 1 (malignancies), and TNF-α was not elevated at all in any of our patients with persistent borreliosis (Figure 2, Table 2). However, Yrjänäinen showed that blocking TNF leads to exacerbation of borreliosis in mice [55]. Overall, studying the regulation and function of TNF-α in humans with borreliosis needs further investigation, also in the context of TNF-α for a more dramatic Jarisch–Herxheimer reaction [56]. Determination of a set of pro-inflammatory cytokines and reduced amounts of anti-inflammatory monocytes/macrophages may constitute a biomarker profile indicating persistent Lyme disease even though the direct identification and quantification of live pathogens by culture and molecular approaches would be more favorable, as shown by a recent study [57]. Correlating biomarker profiles with the molecular identification of persistent *Borrelia* spirochetes [57] despite antibiotic treatment would not only be of prime relevance for further research and diagnostics. Along these lines, we may expand our current understanding of immune regulatory failure in chronic and treatment resistant borreliosis.

Since immunity is further guided by genetic polymorphisms, one approach may be to study functional polymorphisms of inflammatory biomarkers. One of those was addressed in this current study focusing on the IL-6 receptor promotor polymorphism rs1800795. Genotypes of the IL-6 promotor SNP guide the secretion of IL-6, but this effect appeared to be cell type-specific [58]. Accordingly, fibroblast-derived IL-6 may be of high relevance for chronic borreliosis due to the pathogen’s affinity for collagenous tissues, a fact further supported by multiple clinical disease phenotypes [59].

The most significant alterations in the inflammatory cytokine biomarker profile including IL-8 (and IL-6 in individual patients) as well as decreased CD163, were detected in individuals with the heterozygous genotype of the IL-6 promotor SNP (Table 5). Interestingly, the most significant inflammatory marker was IL-8, a cytokine guided by NF-κB activation. A similar observation of increased inflammatory cytokines has been made by us in another patient cohort suffering from chronic pain syndromes with no evidence of infectious complications: Heterozygous individuals (G/C) had higher IL-1β than homozygous G/G or C/C patients (data not shown). In the end, such observations require an appropriately designed case-control setup for validation.

The promotor SNP of IL-6 is rich in CpG islands implying regulation by methylation events, which are known to be altered by chronic inflammatory stress [60]. Nile and coworkers reported increased CpG site methylation and its relationship to mRNA levels in rheumatoid arthritis [61] emphasizing further research along these lines. Overall, sufficient fine tuning of IL-6 signaling appeared to be an important aspect in the correction of immune dysfunction such as chronic inflammation as well as aging. Recently, methylation and acetylation events, involved in aging, have been demonstrated to be sensitive to changes linked to “trained immunity”. Thus chronic inflammation as well as aging related DNA modification could be combatted by chromatin remodeling in innate immune effector cells [62]. Currently, innate immunity is an important target to be studied in patients with persistent Lyme borreliosis such as those who benefit from plant antibiotics as well as compound antioxidant nutritional supplements [63].

## 4. Materials and Methods

### 4.1. Ethics

Patients were analyzed, immune phenotyped and sequenced in course of the routine diagnostic work up. All analyses of biomarkers, flow cytometric studies, and pyrosequencing were performed on request by the caring physicians to investigate hyperinflammation and immune dysfunction in every individual patient. All patients signed a form to agree with isolation of genomic DNA for sequencing of functional SNPs related to inflammation. All subjects gave their informed consent for inclusion before they participated in the study. Investigations were carried out following the rules of the Declaration of Helsinki of 1975 (https://www.wma.net/what-we-do/medical-ethics/declaration-of-helsinki/), revised in 2013. All subjects gave their informed consent for inclusion before they participated in the study. Blood samples were pseudonymized when arriving in the research laboratory and all analyses were performed independently of any personal data. Individual consent to participate in this study was given to the caring physician taking the blood samples.

### 4.2. Patients and Controls

Patient samples were recruited by the Swiss Medical Clinic, Kilchberg, Switzerland and a general practitioner for internal medicine in Pforzheim, Germany. Both institutions are experts in the diagnostic workup and treatment of Lyme borreliosis and related infections and diseases. Humoral immunity against *Borrelia* and coinfections was performed by Gerald Czech-Schmidt (Chronikerlabor, Quedlinburg, Germany). Healthy controls (*n* = 84) were volunteer donors of Ulm University Hospital, Germany. All analyses of biomarkers, flow cytometric studies and pyrosequencing were performed on request by the caring physicians to investigate hyperinflammation and immune dysfunction in every individual patient. All patients signed a form to agree with the isolation of genomic DNA for sequencing of functional SNPs related to inflammation. The analysis was conducted with 2.7 ml of ethylenediaminetetra-acidic acid (EDTA) anti-coagulated blood and 2 ml of heparin anti-coagulated blood for TNF-α measurements.

### 4.3. Flow Cytometry for Phenotyping Patient PBMCs

For cell surface staining, cells were stained for 20 min using anti-CD163-FITC (Ref. 130-097-626, Clone GHI/61.1, Miltenyi Biotec, Bergisch Gladbach, Germany.) and BD Simultest™ Leucogate™ (CD45/CD14; Catalog No. 342408, Clone 2D1/MφP9, BD Biosciences, Franklin Lakes, NJ, USA). Erythrocytes were lysed using FACS Lysing Solution (BD Biosciences, Franklin Lakes, NJ, USA), followed by two washing steps with 1× DPBS (Sigma-Aldrich, St. Louis, MO, USA) and fixed with BD CellFix (BD Biosciences, Franklin Lakes, NJ, USA). Ten thousands cells were counted with a FACSCalibur (BD Biosciences, Franklin Lakes, NJ, USA). Evaluation was performed using Cellquest^TM^ software (BD Biosciences, Franklin Lakes, NJ, USA). Mean fluorescence intensity (MFI) values were mean values of expression densities in the positive population. (x-mean for CD163-FITC or y-mean for CD14-PE).

Healthy controls (CD14: *n* = 75, CD163: *n* = 71) were recruited in Ulm, Germany.

### 4.4. Plasma Biomarkers

Plasma from patients’ blood and control subjects was used for the quantification of biomarkers. A highly sensitive and validated ELISA (Immulite 1000^®^, Siemens, Munich, Germany) was used. The following Immulite 1000^®^ kits were used: I1b to quantify IL-1β (*n* = 106), I6 to quantify IL-6 (*n* = 105), I8 to quantify IL-8 (*n* = 98), and TNA to quantify TNF-α (*n* = 105).

Healthy controls (IL-1β: *n* = 53, IL-6: *n* = 52, IL-8: *n* = 45, and TNF-α: *n* = 54) were recruited in Ulm, Germany.

### 4.5. DNA Isolation

Genomic DNA (gDNA) was isolated from EDTA-blood. The Maxwell 16 LEV Blood DNA Kit (Promega, Madison, WI, USA) was used for automated gDNA isolation Before addition of 30 µL of Proteinase K to 300 µL of the sample, the blood was mixed for approximately five minutes, especially when using thawed material. Then, 300 µL of a lysis buffer, which was supplied in the Kit (AS1290, Promega, Madison, WI, USA), was added. The samples were mixed for about 10 seconds and transferred to a water bath, which had reached a temperature of 56 °C in order to denature proteins and lyse the nuclei. After 20 minutes of incubation, transfer the whole batch to the provided cartridge (AS1290, Promega, Madison, WI, USA). For quantification of the DNA a photometric measurement at a wavelength of 260 nm and 280 nm was performed (Nanodrop^®^ nd-1000, Thermo Fisher Scientific, Waltham, MA, USA). For further experiments DNA was diluted with nuclease-free water to obtain a total amount of 10 ng.

### 4.6. Pyrosequencing

Lyme borreliosis patients (*n* = 107) were genotyped for the IL-6 rs1800795 SNP genotype. PCR was performed using PyroMark^®^ PCR Kit (Cat No. 978703, Qiagen, Hilden, Germany) and Biometra Professional Thermocycler (analytik-jena.de). Following primer pair (biomers.net) was used for amplification: 5′-biotin-TAAGCTGCACTTTTCCCCCTAGTT-3′ (forward, 10 µM), 5′-ATTGTGCAATGTGACGTCCTTTAG-3′ (reverse, 10 µM). Amplification was executed in 40 cycles with an annealing temperature of 57.0 °C.

Pyrosequencing was performed using PyroMark^®^ Gold Q96 Reagents (Cat. No. 972804, Qiagen, Hilden, Germany), PyroMark^®^ Binding Buffer (Cat No. 979006, Qiagen, Hilden, Germany), PyroMark^®^ Annealing Buffer (Cat. No. 979009, Qiagen, Hilden, Germany), Streptavidin–Sepharose Beads (Cat. No. GE17-5113-01, Sigma-Aldrich, St. Louis, MO, USA), and sequencing primer (5′-GTGACGTCCTTTAGCAT-3′, 10 µM; biomers.net). The PCR product was incubated with binding buffer and Streptavidin–Sepharose Beads in a 96 well plate for 10 min under shaking at 1400 rpm in a Thermocycler (room temperature). The double-strand PCR product was then denatured to isolate sepharose-attached single-strands using the vacuum prep tool (Qiagen, Hilden, Germany). Single-strand sequencing was performed by Pyrosequencing using the PSQ HS96 device (Qiagen, Hilden, Germany).

Genotypes of the Control Groups were taken from the *e!Ensembl* Human database (https://www.ensembl.org/Homo_sapiens/Variation/Population?db=core;r=7:22726526-22727526;v=rs1800795;vdb=variation;vf=415970384) (as at July 25, 2019).

### 4.7. Statistical Analysis

Statistical analysis was performed using GraphPadPrism v7 (Graphpad.com). The significance of the genotypes was determined using a Chi-squared test (Pearson’s asymptotic test). The significance of the cytokine plasma concentrations and cell surface markers were calculated by Mann–Whitney-U non-parametric test.

## 5. Conclusions

This study described a novel biomarker profile in persistent Lyme borreliosis patients and a selective preference in patients with selected secondary diseases. The pro-inflammatory mediators IL-1β and IL-8, as well as reduced anti-inflammatory CD163+ macrophages were the most significant parameters. Additionally, the heterozygous G/C genotype of IL-6 promotor SNP rs1800795 turned out to be a promising genetic condition to explain chronic inflammation.

## Figures and Tables

**Figure 1 antibiotics-08-00107-f001:**
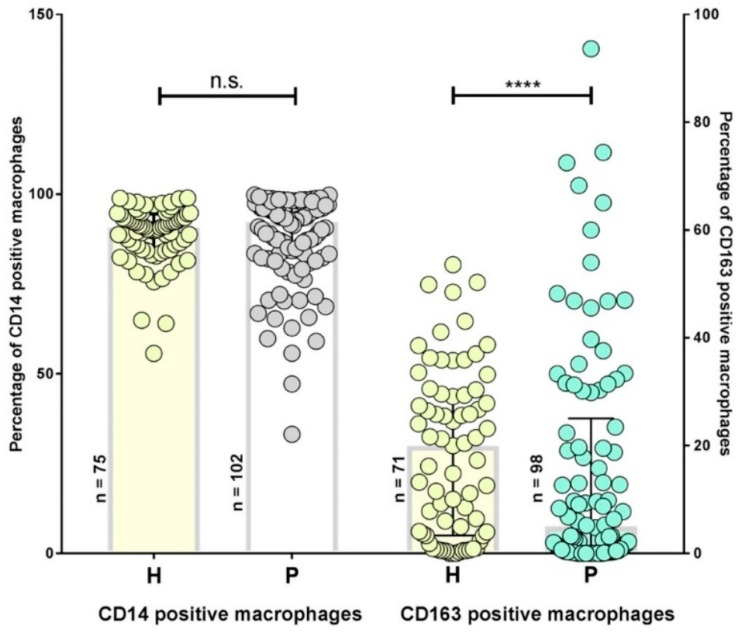
CD14 and CD163 positive monocytes of patients with persistent Lyme disease (P) and healthy controls (H), x-axis. CD14 and CD163 expression was determined by flow cytometry, % positive monocytes are given on the y-axis for both P and H (x-axis). Significance was determined using the Mann–Whitney-U non-parametric test. The CD14 expression in P (median: 92.16%) and H (median: 90.67%; *p* = 0.37); CD163 expression in P (5%) and H (29.96%; *p* < 0.0001). *p*-values > 0.05 (n.s.), ≤ 0.05 (*), ≤ 0.01 (**), ≤ 0.001 (***), ≤ 0.0001 (****; Mann–Whitney-U-test).

**Figure 2 antibiotics-08-00107-f002:**
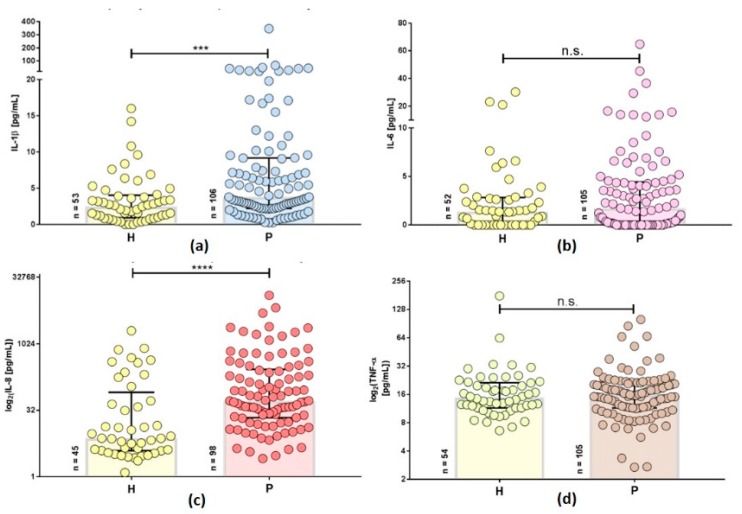
Inflammatory cytokine expression profiles in plasma samples of patients with persistent Lyme borreliosis (P) and healthy controls (H), x-axis. IL-1β and IL-6 concentrations (pg/mL), IL-8 and TNF-α concentrations (log_2_(pg/mL)) are shown on the left y-axis. Mann-Whitney-U non-parametric test was used to determine the statistical significance. n.s. indicates a *p*-value > 0.05, (***) indicates a *p*-value ≤ 0.001, (****) indicates a *p*-value ≤ 0.0001. (**a**) IL-1β in P (median: 3.785 pg/mL) and H (median: 2.48 pg/mL). (**b**) IL-6 in P (median: 1.74 pg/mL) and H (median: 1.375 pg/mL). (**c**) IL-8 in P (median: 52.4 pg/mL) and H (median: 7.45 pg/mL) (**d**). TNF-α in P (median: 15.9 pg/mL) and H (median: 15.2 pg/mL)*. p*-values > 0.05 (n.s.), ≤ 0.05 (*), ≤ 0.01 (**), ≤ 0.001 (***), ≤ 0.0001 (****; Mann–Whitney-U-test).

**Table 1 antibiotics-08-00107-t001:** Distribution of the patients into four subpopulations with a pattern of secondary disease phenotypes. First group consists of patients with malignancies, group 2 of patients with an additional autoimmune diseases (including e.g., multiple sclerosis, arthritis, diabetes mellitus type I, Hashimoto thyroiditis, and Morbus Basedow), group 3 of patients with additional neurological dysfunctions (including Parkinson’s disease (PD), amyotrophic lateral sclerosis (ALS), chronic fatigue syndrome (CFS), and depression), and group 4 of patients with unspecific clinical symptoms but multiple bacterial and/or viral coinfections. All groups (total) have increased IL-8, and TNF- α is not different from controls (see table below).

Group *	Secondary Disease	Patient Number	Inflammatory Phenotype
1	Malignancies	10	IL-1β↑, IL-6↑, IL-8↑ CD163+↓ TNF-α*
2	Autoimmunity	27	IL-8↑, CD163+↓ TNF-α*
3	Neurological dysfunctions (including PD, ALS, psychiatric diseases)	30	IL-1β↑, IL-8↑ TNF-α*
4	Multiple infections	40	IL-1β↑, IL-8↑, CD163+↓ TNF-α*
**1–4**	**Total**	107	IL-8↑, TNF-α *

* TNF-α concentrations were not different in any disease groups linked to persistent Lyme borreliosis.

**Table 2 antibiotics-08-00107-t002:** Cytokine expression or percentages of CD163 positive monocytes/macrophages related to secondary diseases (1–4), as compared to healthy controls. Median values of IL-1β, IL-6, IL-8, and TNF- α (pg/mL), medians of CD163+ macrophages (% positive). *p*-values > 0.05 (n.s.), ≤ 0.05 (*), ≤ 0.01 (**), ≤ 0.001 (***), ≤ 0.0001 (****; Mann–Whitney-U-test).

Mediator	Group (Secondary Diseases)	Patient Groups	Median (±Range)	Significance (Groups *vs.* Control)
IL-1β	1	Malignancies	6.03 (±64.8)	**
	2	Autoimmunity	2.88 (±21)	n.s.
	3	Neurological Dysfunctions	3.28 (±44.4)	**
	4	None	5.69 (±346.4)	**
	Ctrl	Control	2.48 (±16)	
IL-6	1	Malignancies	4.11 (±36.5)	*
	2	Autoimmunity	1.4 (±13.9)	n.s.
	3	Neurological Dysfunctions	0.76 (±45.3)	n.s.
	4	None	2.08 (±64.7)	n.s.
	Ctrl	Control	1.38 (±30.2)	
IL-8	1	Malignancies	406 (±5013.2)	***
	2	Autoimmunity	37.2 (±2370.8)	**
	3	Neurological Dysfunctions	60 (±12726)	**
	4	None	49 (±6656.4)	**
	Ctrl	Control	7.45 (±2011.7)	
TNF-α	1	Malignancies	19.7 (±35.4)	n.s.
	2	Autoimmunity	15.2 (±25.1)	n.s.
	3	Neurological Dysfunctions	15.4 (±58.8)	n.s.
	4	None	17.7 (±97.3)	n.s.
	Ctrl	Control	14.8 (±171.4)	
CD163+ Macrophages	1	Malignancies	1.71 (±93.6)	*
	2	Autoimmunity	2.98 (±65)	***
	3	Neurological Dysfunctions	9.7 (±74.4)	n.s.
	4	None	5.63 (±68.3)	**
	Ctrl	Control	29.96 (±84.4)	

**Table 3 antibiotics-08-00107-t003:** Frequencies of IL-6 rs1800795 genotypes in Lyme borreliosis patients compared to a reference cohort (e!Ensembl)* by two-df (degrees of freedom) Chi-square test, (*) reference genotypes: [47].

SNP Genotype	Lyme Borreliosis Patients *n* (%)	*E!Ensembl* Reference Population *n* (%)
IL-6 rs1800795 Genotype		
G/G	36 (33.6%)	181 (36%)
G/C	59 (55.1%)	226 (44.9%)
C/C	12 (11.2%)	96 (19.1%)
**Total**	**107 (100%)**	**503 (100%)**

*p* = 0.075 (n.s.) **χ**^2^ = 5.191.

**Table 4 antibiotics-08-00107-t004:** Frequencies of the IL-6 promotor rs1800795 genotype in Lyme borreliosis patients according to group designations of secondary disease phenotypes (Group 1–4). *p*-values from two-df Chi-square test.

Secondary Disease Group	Malignancies n (%) 1	Autoimmunity n (%) 2	Neurological Dysfunction n (%) 3	None * n (%) 4
IL-6 rs1800795				
G/G	0 (0%)	7 (25.9%)	13 (43.3%)	16 (40%)
G/C	7 (70%)	18 (66.7%)	13 (43.3%)	21 (52.5%)
C/C	3 (30%)	2 (7.4%)	4 (13.3%)	3 (7.5%)
**Total**	**10 (100%)**	**27 (100%)**	**30 (100%)**	**40 (100%)**
p =	0.0620	0.089	0.6264	0.1864
χ^2^ =	5.56	5.548	0.9355	3.359

* The patient group designated as “None” was characterized by multiple infectious complications.

**Table 5 antibiotics-08-00107-t005:** Cytokine plasma concentrations and percentages of CD163 positive M2 macrophages related to the IL-6 promotor SNP rs1800795 genotypes. Mann–Whitney-U non-parametric tests were applied. *p*-values > 0.05 (n.s.), ≤ 0.05 (*), ≤ 0.01 (**), ≤ 0.001 (***), ≤ 0.0001 (****).

Mediator	Genotype IL-6 Promotor rs1800795	Median (±Range)	Significance (Genotype vs. Control)
IL-1β	G/G	3.27 (±346.4)	*
G/C	3.52 (±65.6)	**
C/C	5.81 (±18.7)	**
Control	2.48 (±16)	
IL-6	G/G	0.3 (±7.6)	n.s.
G/C	2.88 (±64.7)	**
C/C	0.12 (±29.3)	n.s.
Control	1.38 (±30.2)	
IL-8	G/G	39.3 (±12723.4)	*
G/C	66 (±5016.3)	****
C/C	168 (±1220.8)	**
Control	7.45 (±2011.7)	
CD163+Macrophages	G/G	3.28 (±72.49)	**
G/C	3.93 (±68.26)	****
C/C	12.66 (±93.64)	n.s.
Control	29.96 (±84.4)

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
