# Peer review of "Cytokine Expression Patterns and Single Nucleotide Polymorphisms (SNPs) in Patients with Chronic Borreliosis"

_antibiotics, 2019, doi:10.3390/antibiotics8030107_

Round 1

Reviewer 1 Report

Cytokine expression patterns and single nucleotide polymorphisms(SNPs) in patients with chronic borreliosis

As an increasing prevalent and global infectious disease, Lyme borreliosis has successfully drawn people’s attention. In this manuscript, the authors studied the changes of macrophages, inflammatory biomarkers and SNP of IL-6 promoter in Lyme borreliosis patients. In a summary, the authors demonstrated the decrease of CD163+ macrophage, up-regulated IL-1βand IL-8, as well as the enrichment of C/G phenotype in IL-6 promoter of Lyme borreliosis patients, which may provide some cues for the mechanism study of inflammatory phenotype in Lyme borreliosis patients and future treatment for Lyme borreliosis disease

Overall the manuscript showed clinical inflammatory phenotype, as well as the SNP of IL-6 promoter in a large patient cohort. Nevertheless, there is no mechanism study of these clinical symptoms and the correlation between the inflammation and SNP of IL-6. The study used statistics methods to reveal the inflammation phenotype and SNPs based on human patient samples. I may recommend its publication in Antibiotics with some major modifications.

Major points:

1.    The authors may need to have some explanation on the use of Mann-Whitney-U non-parametric tests in the whole manuscript. The p-values is the only indicator for the significance between patient samples, even the samples may have a very large range. To make the most appropriate conclusion, would it be necessary to double-check by another test method?

2.    In table5, the control healthy group should have different genotype (GG, CG, CC), which may show related cytokine plasma concentration and CD163+ population. So when comparing GG genotype in patient with control group, the genotype of control should be GG also.

3.    Actually the cytokine expression of IL6 is not changed between the Lyme borreliosis patients and healthy group, even in different secondary disease groups (group 1-4, only slight significant in Group 1 Malignancies), why the authors still put emphasis on the promoter of IL-6 instead of cytokine IL-1βand IL-8?

Minor points:

1.    Figure 1 & 2 each dot represents single patient? If yes, put the number there. Also, showing some image representatives of flow cytometry data in figure 1 would be much more intuitive.

2.    Having some represented pyrosequencing data for the demonstration of IL-6 RS1800795 genotype would be better.

3.    Table 3 & 4, to make it comparable, it would be better to put the percentage of each genotype instead of only patient number. 

4.    Table 5, just distinguish “,” from”.” in the numbers……. It totally means different.

5.    Line 156, highlighted Table 2should be Table 3?

6.    Line 138, “Patients with malignancies (Group 1) did notpresent with significantly increased……” Is ”NOT” ?? I think it should be did present.

I suggest to have another last paragraph showing its significance of this study, as well as few sentence in the end of introduction.

Author Response

Reviewer 1

Major points:

1. The authors may need to have some explanation on the use of Mann-Whitney-U non-parametric tests in the whole manuscript. The p-values is the only indicator for the significance between patient samples, even the samples may have a very large range. To make the most appropriate conclusion, would it be necessary to double-check by another test method?

Answer to the reviewer: Thank you for this important argument. The Mann-Whitney U test (MWU) was chosen to compare two samples  with regard to the location of continuous traits with non-normal distributions. However, this test may also be, to a certain degree, responsive to differences in dispersion ("omnibus test"). Therefore, we additionally performed two-sample t-tests based on the Satterthwaite approach for unequal variances. In the case of percentages we first performed an arcsine-square-root transformation of the observed values, in the case of CD14 and CD163 expression density determinations, we applied the T-test after log-transformation.

MWU and T-tests yielded only slightly deviating results. All essential conclusions of the study were definitely confirmed by both tests.  The authors appreciate effort and recalculation of our statistical results by Josef Hoegel. All co-authors agreed to co-authorship in the new version of the manuscript.

>> Tabea M. Hein 1, Philip Sander 1, Anwar Giryes 2, Jan-Olaf Reinhardt 3, Josef Hoegel 4, and E. Marion Schneider 1,*

1 Division of Experimental Anesthesiology, University Hospital Ulm, Albert-Einstein-Allee 23, 89081 Ulm, Germany

2 Swiss Medical Clinic AG, Grütstrasse 55, CH-8802 Kilchberg, Switzerland

3   Practice for General Medicine, Sports Medicine, and Endocrinology, Leopoldstraße 17, 75172 Pforzheim

  4     Institute of Human Genetics, Ulm University, Albert-Einstein-Allee 11, Ulm, D-89081, Germany <<

2. In table5, the control healthy group should have different genotype (GG, CG, CC), which may show related cytokine plasma concentration and CD163+ population. So when comparing GG genotype in patient with control group, the genotype of control should be GG also.

Indeed this is an important argument which we currently addressed this context for another inflammatory disease i.e. patients suffering from chronic pain syndromes, but had no evidence for infectious complications. When comparing the heterozygous genotype (C/G) with homozygous wildtype (G/G) and homozygous mutated (C/C) we found increased IL-1in heterozygous (C/G) individuals when compared to homozygous mutated (C/C) as well as homozygous wildtype (G/G). To confirm the trend of increased inflammatory cytokines in heterozygous patients need to be addressed in a clinical study designed in a case-control format. To explain this context in greater detail the following sentence has been inserted into the discussion of our manuscript:

>> Interestingly the most significant inflammatory marker was IL-8, a cytokine guided by NF-B activation. A similar observation of increased inflammatory cytokines has been made in another patient cohort suffering from chronic pain diseases in the absence of infectious complications: Heterozygous individuals (C/G) had higher IL-1 than homozygous G/G or C/C patients (data not shown). In the end, these observations require an appropriately designed case-control setup for validation. A correlation with additional disease phenotypes may be more noteworthy in a larger patient cohort. <<

3. Actually the cytokine expression of IL6 is not changed between the Lyme borreliosis patients and healthy group, even in different secondary disease groups (group 1-4, only slight significant in Group 1 Malignancies), why the authors still put emphasis on the promoter of IL-6 instead of cytokine IL-1βand IL-8?

We addressed the potential relevance of IL-6 in pathogen-associated diseases such as chronic, treatment-resistant borreliosis, since the IL-6 promotor polymorphism rs1800795 plays a major role in i) inflammation of immune cells [39, 40], ii) antibody production (Amr et al. 2016), including various autoimmune diseases (Tanaka et al. 2014) and iii) type 2 diabetes [42], caused by multifactorial metabolic dysregulation. On a secondary level, the IL-6 genotype also influences other inflammatory markers including IL-1 IL-8, and numbers of circulating CD163+ monocytes. All these biomarkers may play a role in the pathogenicity of chronic borreliosis. To better explain why we concentrated on IL-6 and its promotor polymorphism, we inserted the following text into the introduction of our manuscript:

>> […], however significant improvement has been described for patients suffering from chronic Lyme borreliosis by antioxidative treatment regimen (Peacock et al. 2015).<<

>> IL-6 is unique by its properties as an inflammatory as well as regulatory cytokine influencing IL-1, IL-8 and TNF-Schindler et al. 1990).<<

Amr K, El-Awady R, Raslan H. Assessment of the -174G/C (rs1800795) and -572G/C

(rs1800796) Interleukin 6 Gene Polymorphisms in Egyptian Patients with Rheumatoid

Arthritis. Open Access Maced J Med Sci. 2016 Dec 15;4(4):574-577. doi:

10.3889/oamjms.2016.110. Epub 2016 Oct 11. PubMed PMID: 28028393; PubMed Central

PMCID: PMC5175501.

Peacock BN, Gherezghiher TB, Hilario JD, Kellermann GH. New insights into Lyme disease. Redox Biol. 2015;5:66–70. doi:10.1016/j.redox.2015.03.002

Tanaka T, Narazaki M, Ogata A, Kishimoto T. A new era for the treatment of inflammatory autoimmune diseases by interleukin-6 blockade strategy. Semin Immunol. 2014 Feb;26(1):88-96. doi: 10.1016/j.smim.2014.01.009. Epub 2014 Mar 1. Review. PubMed PMID: 24594001.

Minor points:

1.    Figure 1 & 2 each dot represents single patient? If yes, put the number there. Also, showing some image representatives of flow cytometry data in figure 1 would be much more intuitive.

In our revised version, we improved Figure 1 and Figure 2 accordingly, and also inserted the number of individuals tested as requested. Moreover we added supplementary Figure S1 to explain gating strategies for the identification of CD163+  monocytes in whole blood isolates. The newly inserted text is as follows:

>> In whole blood samples we identified monocytes by i) scatter analysis and ii) expression densities of CD45 and CD14. Monocytes identified by scatter analysis, were then tested for expression of FITC-labeled anti-CD163. Gating strategies are explained in supplementary Figure S1. As demonstrated in Figure 1, CD14 expression is not different between healthy donors and persistent Lyme borreliosis patients (p=0.37). However, the amount of CD163 positive monocytes was much lower in patients as compared to healthy controls (Figure 1). <<

2.    Having some represented pyrosequencing data for the demonstration of IL-6 RS1800795 genotype would be better.

We thank the reviewer for the option to present a representative example of our pyrograms which are unequivocal for the identification of SNPs. Therefore, supplementary Figure S2 has been introduced into the manuscript and the following text has been added:

>> A representative pyrogram for the detection of this SNP in wildtype, heterozygous and homozygously mutated gDNA is shown in supplementary Figure S2. <<

3. Table 3 & 4, to make it comparable, it would be better to put the percentage of each genotype instead of only patient number.

Thank you very much for raising this argument. We know inserted patient numbers in addition to the percentage of genotypes into Table 3 and Table 4. For Table 4 we inserted a better description of patients designated "None" into the legend:

>> * the patient group designated as "None" was characterized by multiple infectious complications. <<

4.    Table 5, just distinguish "," from"." in the numbers……. It totally means different.

We now corrected the punctuation in Table 5 and also corrected Chi-square and the p-value for group 2 (patients with autoimmune disease). The updated Chi square and p-value were 5.548 and p=0.089, respectively.

5.    Line 156, highlighted Table 2should be Table 3?

Thank you very much. This mistake has been corrected in the revised version of our manuscript.

6.    Line 138, "Patients with malignancies (Group 1) did notpresent with significantly increased……" Is "NOT" ?? I think it should be did present.

The reviewer is fully correct in that group 1 displayed significantly increased cytokine levels. This context has been corrected in the new version of the manuscript:

>> Patients with malignancies (Group 1) showed increased IL-1, IL-6 and IL-8 as well as with decreased numbers of CD163+  M2 macrophages. <<

I suggest to have another last paragraph showing its significance of this study, as well as few sentence in the end of introduction.

Thank you very much for this important notice. The authors decided to include hypothetical ideas addressing the context of immune insufficiencies related to chronic inflammation and IL-6 in greater detail. In the Introduction, the following paragraph has been inserted:

>> IL-6 is an unique cytokine by its potency to upregulate inflammation as well as anti-inflammatory and tissue reconstitutive pathways and antibody-production. These opposing functions of IL-6 depend on IL-6 receptors (p80 and gp130), expressed by different target tissues, and function as IL-6 buffers [45]. Basically, persistent Lyme borreliosis may occur by a detrimental effect of chronic inflammation as well as specific or generalized immune insufficiencies. This context motivated us to concentrate on genetics of the IL-6 production but nevertheless also determined IL-1, IL-8 and TNF-on the protein level. <<

Correspondingly, a new paragraph was added to the discussion:

>> Overall, sufficient fine tuning of IL-6 signaling appeared to be an important aspect in the correction of immune dysfunction such as chronic inflammation as well as aging. Recently, methylation and acetylation events, involved in aging, have been demonstrated to be sensitive to changes linked to "trained immunity". Thus chronic inflammation as well as aging related DNA modification could be combatted by chromatin remodeling in innate immune effector cells . Currently, innate immunity is an important target to be studied in patients with persistent Lyme borreliosis who benefit from plant antibiotics as well as compound antioxidant nutritional supplements .  <<

Reviewer 2 Report

This is a highly significant and well researched article and needs to be published. It will be a major contribution to our knowledge of this complex infectious disease. 

I have three comments for minor corrections.

Line 42: Although the authors are quoting the wording of these sources, the symptoms listed are not just subjective. They can be further evaluated with objective testing. Suggest deleting the word subjective.

Line 81: Why is the Method section after the Results? Suggest moving the Method section before the Results section.

Line 102: Footnote 57 out of sequence, was [57] supposed to be [37]? Looks like this was a typographical error. 

Author Response

Reviewer 2:

This is a highly significant and well researched article and needs to be published. It will be a major contribution to our knowledge of this complex infectious disease.

I have three comments for minor corrections.

Line 42: Although the authors are quoting the wording of these sources, the symptoms listed are not just subjective. They can be further evaluated with objective testing. Suggest deleting the word subjective.

The authors thank the reviewer for bringing up this issue, which we now emphasized as follows:

>>[…] ~10% of the patients report a persistence of disease-relevant symptoms like fatigue […] <<

Line 81: Why is the Method section after the Results? Suggest moving the Method section before the Results section.

The template form of Antibiotics implies the description of Material and Methods following the discussion. The authors just adapted their ms to the given template.

Line 102: Footnote 57 out of sequence, was [57] supposed to be [37]? Looks like this was a typographical error.

Thank you very much for pointing out the mistaken reference number. In the new version of the manuscript, this mistake has been corrected.

Reviewer 3 Report

Minor comments: It will be more strong if the author can increase the n and reach to n=40. 

Author Response

Reviewer 3

Minor comments: It will be more strong if the author can increase the n and reach to n=40.

The reviewer is fully right: An increased number of patients and controls investigated would result in more reliable significances of our results. For the time being we concentrated on appropriate diagnostic criteria for persistent Lyme borreliosis. To increase the study cohort as well as other SNPs guiding inflammatory cytokine responsiveness are in progress.

Round 2

Reviewer 1 Report

The manuscript has been significantly improved after revision. I recommend its publication in Antibiotic Journal.